# Living-Cell Diffracted X-ray Tracking Analysis Confirmed Internal Salt Bridge Is Critical for Ligand-Induced Twisting Motion of Serotonin Receptors

**DOI:** 10.3390/ijms22105285

**Published:** 2021-05-17

**Authors:** Kazuhiro Mio, Shoko Fujimura, Masaki Ishihara, Masahiro Kuramochi, Hiroshi Sekiguchi, Tai Kubo, Yuji C. Sasaki

**Affiliations:** 1AIST-UTokyo Advanced Operando-Measurement Technology Open Innovation Laboratory (OPERANDO-OIL), National Institute of Advanced Industrial Science and Technology (AIST), 6-2-3 Kashiwanoha, Chiba 277-0882, Japan; shoko-san@aist.go.jp (S.F.); masaki.ishihara30@gmail.com (M.I.); masahiro-kuramochi@edu.k.u-tokyo.ac.jp (M.K.); tai.kubo@edu.k.u-tokyo.ac.jp (T.K.); 2Graduate School of Frontier Sciences, The University of Tokyo, 5-1-5 Kashiwanoha, Chiba 277-8561, Japan; 3Center for Synchrotron Radiation Research, Japan Synchrotron Radiation Research Institute, 1-1-1 Kouto, Sayo-cho, Hyogo 679-5198, Japan; sekiguchi@spring8.or.jp

**Keywords:** GPCR, 5-HT_2A_ receptor, intramolecular dynamics, live cell, X-ray diffraction analysis

## Abstract

Serotonin receptors play important roles in neuronal excitation, emotion, platelet aggregation, and vasoconstriction. The serotonin receptor subtype 2A (5-HT_2A_R) is a Gq-coupled GPCR, which activate phospholipase C. Although the structures and functions of 5-HT_2A_Rs have been well studied, little has been known about their real-time dynamics. In this study, we analyzed the intramolecular motion of the 5-HT_2A_R in living cells using the diffracted X-ray tracking (DXT) technique. The DXT is a very precise single-molecular analytical technique, which tracks diffraction spots from the gold nanocrystals labeled on the protein surface. Trajectory analysis provides insight into protein dynamics. The 5-HT_2A_Rs were transiently expressed in HEK 293 cells, and the gold nanocrystals were attached to the N-terminal introduced FLAG-tag via anti-FLAG antibodies. The motions were recorded with a frame rate of 100 μs per frame. A lifetime filtering technique demonstrated that the unliganded receptors contain high mobility population with clockwise twisting. This rotation was, however, abolished by either a full agonist α-methylserotonin or an inverse agonist ketanserin. Mutation analysis revealed that the “ionic lock” between the DRY motif in the third transmembrane segment and a negatively charged residue of the sixth transmembrane segment is essential for the torsional motion at the N-terminus of the receptor.

## 1. Introduction

G-protein-coupled receptors (GPCRs) play important roles in a wide variety of physiological functions, such as signal sensing, immune system processes, and neurotransmission. Metabotropic serotonin receptor subtype 2A (5-HT_2A_R) is widely distributed in the central nervous system, and are involved in neuronal excitation, learning, and emotion. The 5-HT_2A_Rs are also important for platelet aggregation, vasoconstriction, and vascular smooth muscle proliferation [1,2,3]. The activation of Gq-coupled 5-HT_2A_Rs results in activation of phospholipase C, which consequently hydrolyzes phosphatidylinositol 4,5-bisphosphate into secondary messengers’ inositol 1,4,5-trisphosphate and diacylglycerol [4].

The structures of 5-HT_2A_R have been solved by crystallography and cryoelectron microscopy [5,6]. However, it is technically challenging to detect the intramolecular motions of membrane proteins directly, because signal transduction comprises a multi-step movement with various directions and speed ranging from a microsecond (μs) to a millisecond (ms) time scale. They are beyond the temporal and spatial resolution of most imaging-based techniques.

To elucidate the fast and microscale dynamics of 5-HT_2A_Rs in live cells, we used the diffracted X-ray tracking (DXT) technique in this analysis. The DXT analyzes the motion of target proteins through statistical analysis of diffraction spots from the gold nanocrystals labeled on the protein surface [7,8,9,10]. Using a synchrotron radiation source and a high-speed detector, conformational dynamics, as well as functional intermediates, can be detected in a time-resolved manner from μs to ms scale with sub-angstrom spatial information. We have also recently developed a diffracted X-ray blinking (DXB) technique, which uses low-dose monochromatic X-rays and analyzes continuation of signals by calculating the autocorrelation function of each pixel from the recorded data [11,12].

In our previous study using DXT and DXB techniques, we were able to detect various 5-HT_2A_Rs motion components according to the recording frame rate [13]. In that, the ligand-induced conformational shift from multiple structures to active or inactive conformation at different time scales was observed. However, the detail in the twisting movement of each transmembrane segment was not clarified yet. Using the lifetime filtering technique [10], we analyzed here the torsional movement of the N-terminal domain of 5-HT_2A_R.

## 2. Results

We analyzed the ligand induced intramolecular motion of 5-HT_2A_Rs in living cells using the DXT technique (Figure 1a,b). The motion was recorded at a frame rate of 100 μs per frame, and the resulting diffraction spots were analyzed in two rotational axes of tilt (θ) and twist (χ) angles. We analyzed 319 through 1291 trajectory points for each condition.

The FLAG sequence (DYKDDDDK) was introduced genetically at the N-terminal end of the 5-HT_2A_R. The HEK293 cells expressing 5-HT_2A_Rs were cultured on the surface of polyimide films, and its N-terminus was labeled with anti-FLAG antibody conjugated with gold nanocrystals (Figure 1c,d). Expression and function of the expressed receptor constructs were confirmed by the immunocytochemistry, live-cell immunofluorescence assay and calcium influx assay (Appendix A). We also confirmed using the trypan blue dye exclusion method that the cell viability under all experimental conditions was as high as 97% (data not shown). The cells were cultured at confluency so that every recording spot provides effective signals as well as reducing background from the nonspecifically bound golds. As for the twisting motion, the negative value (Δχ < 0) represents the clockwise rotation (CW), while the positive value (Δχ > 0) represents the counter-clockwise rotation (CCW) (Figure 1e).

### 2.1. DXT Analysis of 5-HT_2A_R

In order to apply DXT to living cells, radiation damage should be minimized. In the previous study, we found that narrowing the bandwidth of X-ray eliminated cell damage [13]. So we used X-rays with 0.02 energy bandwidth, 15 keV peak energy and 10^13^ photon/s flux for the living-cell DXT, instead of the 15.8 keV white X-ray that is commonly used for single molecular DXT.

We first analyzed the time-averaged mean square displacement (MSD) curves using the diffraction tracks obtained (Figure 2a and Appendix A) [14]. As reported in the previous study, the slope in the θ direction was suppressed by the full agonist α-methylserotonin (αMS) and the αMS with inverse agonist ketanserin (αMS/ket) (Figure 2a). This was considered due to the shift of conformational equilibrium from multiple structures to active or inactive conformation [13,15].

This time the MSD curves in the χ directions were analyzed (Figure 2a, lower panel). Slopes of αMS andαMS/ket in χ direction were also decreased from that of unliganded 5-HT_2A_R. Although most conditions represent restricted curves (α < 1), the unliganded and αMS were superdiffused (Appendix A, α = 1.67 and 1.74, respectively). For simple Brownian diffusion, MSD plots represent linear slope of 4D, where D is the diffusion coefficient. These superdiffusion plots, however, are indicative of directed diffusion.

To understand the reason of superdiffusion curve of the unliganded receptor, we analyzed the motion of individual receptor. The displacement distribution graphs represent the travel distance from the original point in 1 ms (100 μs × 10 frames) and 1.6 ms (100 μs × 16 frames) (Figure 2b). The MSD curve shows the average motion, while the displacement distribution represents the motion of the constituent components.

In addition to the main population peaking at Δχ = 0, another population representing the torsion movement of CW (Δχ < 0) was observed (Figure 2b, red arrows). This high mobility group was also recognized in the two-axis distribution graphs of θ and χ coordinates (dotted circles in the Figure 2b, lower panels), which comprise 33% of the total probability density at Δt = 1 ms (Appendix A). Subpopulation peaks were −10.1 mrad at Δt = 1 ms, and −19.1 mrad at Δt = 1.6 ms.

The displacement distribution of 10 μM αMS was mostly fitted by single Gaussian curves and the fast moving subpopulation was not observed (Figure 2c). GPCR proteins in the native condition have a conformational balance among inactive, active, and a number of intermediate states. Elimination of high mobility group by αMS may represent the shift of conformational equilibrium from multiple structures to active conformation [15].

Ketanserin is a competitive inverse agonist to 5-HT_2A_R with Ki = 2–3 nM (The PDSP Ki database; https://pdsp.unc.edu/databases/kidb.php (accessed on 1 May 2021). We then analyzed the 5-HT_2A_R motion with αMS/ket. The variance (*σ*^2^) of the Gaussian curves fitted for αMS/ket (1.3 at Δt = 1 ms) was much lower than those of other conditions (2.6 and 2.7 for unliganded and αMS, respectively), suggesting the 5-HT_2A_R treated with αMS/ket appeared to have a reduced fluctuation (Figure 2d and Appendix A).

The intramolecular motion of the wild-type 5-HT_2A_R was further analyzed over time from the two-axis map (Figure 3). The unliganded 5-HT_2A_R clearly shows the highly mobile group with the CW rotation (Δχ < 0), which mostly have the higher θ value (Figure 3a, indicated by dotted circles). The mean value of the Gaussian curves adapted to the highly mobile group was −10.1 mrad (Appendix A). The high mobility group was not present in both αMS and αMS /ket (Figure 3b,c). In response to the smaller variance (*σ*^2^) in the Gaussian curve, the population with low mobility remained in αMS/ket even at Δt = 1.6 ms (Figure 3c, dark filled pixels indicated by a red arrow).

### 2.2. Motion Comparison by Two-Axis Subtraction Maps

To understand the ligand induced dynamics of 5-HT_2A_R, subtraction analysis was performed. In the subtraction map between the unliganded and αMS, the population in fast movement was dominated in unliganded (Figure 4a, red pixels). Most of them belonged to high θ scores, and the torsion direction was CW (Δχ < 0). On the contrary, the subtracted αMS was mainly found at Δχ ~ 0 mrad with low θ scores (Figure 4a, blue pixels). The mobility of 5-HT_2A_R was further reduced by αMS/ket and its large population remains at Δχ ~ 0 mrad (Figure 4b). This is consistent with the presence of αMS/ket low mobility populations (Figure 2d).

The high mobility population of unliganded 5-HT_2A_R was also highlighted by the time series subtraction graphs between the unliganded and αMS (Figure 4c, red arrows).

### 2.3. Motion Analysis of the DRY/AAY Mutant

The displacement distribution for the DRY/AAY mutant was also analyzed. The DRY/AAY mutation disrupts an “ionic lock” between the third transmembrane segment and the second intracellular loop. Because the ionic lock stabilizes the structure in an inactive form, its disruption often undergoes constitutive activity [16,17].

The displacement distribution of the DRY/AAY mutant fitted with a single peak shows the distribution is not symmetrical to the y-axis; shoulder angle at the minus Δχ (left side of the shoulder) was steeper than the plus Δχ (right side of the shoulder) (Figure 5a, upper panels). The motion of DRY/AAY mutant proteins are confined and are considered to have fluctuated asymmetrically along the χ axis. In three Gaussian curve fitting, the main peak position was shifted from χ = 0.2 mrad in the WT (Appendix A-peak2, 60% area ratio at Δt = 1 ms) to χ = −0.9 mrad in the DRY/AAY (Appendix A-peak1, 83% area ratio at Δt = 1 ms), suggesting clear bias of DRY/AAY mutant to the CW direction. This probably represents a conformational shift from structural equilibrium of multiple states of the WT to a stabilized active conformation in DRY/AAY due to the disruption of ionic lock. The fast-moving population was also not found in both αMS and αMS/ket (Figure 5b,c). The two-axis subtraction maps and the time course of subtraction in the χ distribution have not shown a clear difference between the unliganded and αMS in the DRY/AAY mutant (Figure 5d,e).

The two-axis subtraction map between the wild-type (unliganded) and the DRY/AAY (unliganded) again confirmed that the fast-moving population with CW movement (Δχ < 0) is dominated in the wild-type (Figure 5f). A large population of DRY/AAY remains at Δχ ~ 0.

## 3. Discussion

In this study, the dynamics of 5-HT_2A_R were analyzed in living cells using DXT techniques. The bandwidth of X-rays was reduced to eliminate cell damage, compared to the X-rays used in a single DXT molecule. We have successfully detected the torsional movement of 5-HT_2A_R in living cells.

From this analysis, a clockwise torsional motion of 10.1 mrad (0.58°) was elucidated at Δt = 1 ms at the N-terminal end of the unliganded receptor. While DXT is a very sensitive measurement method, the resulting value seemed to be slightly larger than the previous results for the TRPV1 channel (averaged motion of 0.1 mrad at Δt = 1 ms) [10] and the estimation from the structural analysis [6].

In this experiment, the N-terminal end was labelled with a gold nanocrystal. The movement of the N-terminus can include the parallel transition on the plane of transmembrane segment 1 and the spatial arrangement caused by the torsion of the axis.

The N-terminal region of Type A GPCRs has numerous functions in ligand recognition, surface expression and signaling [18]. The N-terminal region of serotonin 2B receptor has a negative modulation function, and its R6G;E42G mutations increase agonist binding, cell proliferation and slow its desensitization kinetics [19]. Structural analysis of 5-HT_2A_R showed that conformational rearrangements generated agonistic binding selectivity. The slow binding kinetics of lysergic acid diethylamide (LSD) to 5-HT_2A_R are driven by a “lid” formed at the binding pocket by extracellular loop 2, which stabilizes the ligand-receptor complex [20]. Indeed, the molecular docking studies support the ligand-induced conformational rearrangement of the serotonin receptors [21,22].

The N-terminal structures of GPCRs type A are not well understood as they are considered intrinsically disordered. Indeed, most structural data do not include information about the N-terminus. The findings of this study show that the movement of N-terminus varies considerably depending on the presence or absence of the ligand, suggesting that the 5-HT_2A_R N-terminal can also have a certain effect on bound ligand stabilization. To separate the motions of the N-terminal domain and of the transmembrane 1, comparative DXT experiments using N-terminally truncated and/or other mutants that disturb the interaction of the N-terminus with the TMs core will be necessary. An overall picture of 5-HT_2A_R motion is expected through detailed time-resolved measurements of each functional domain.

The “ionic lock” between the E/DRY motif of the third transmembrane segment and a negatively charged residue of the sixth transmembrane segment plays an important role in the regulation of the GPCR function [16]. Because the ionic lock stabilizes the structure in an inactive form, its disruption frequently leads to a constitutive active receptor [16,17]. In this study, the mutant DRY/AAY protein which disturbs the ionic lock lost the clockwise torsion motion observed in the wild type. This is consistent with our previous report that the loss of ionic lock lost the fluctuation of the receptor analyzed in the θ direction [13].

In the native state, most GPCRs are considered to have multiple structures under the conformational balance between active and inactive conformations. The rapid-motion group observed in the unliganded receptor histograms can be the population representing the displacement of these two stages. This DXT study confirmed the critical role of the intramolecular ionic lock in switching between active and inactive forms. A recent study using fluorine nuclear magnetic resonance spectroscopy demonstrated adenosine A_2A_ receptors facilitate at least two distinct active states and three inactive states [23]. The number of active conformations introduced by the breakage of ionic lock in DRY/AAY still remains unclear.

In this DXT analysis, we mainly focused on the intramolecular dynamics of 5-HT_2A_R. However, to understand the total regulation of the receptor, intermolecular association with the effector proteins such as G-proteins and arrestins, and higher ordered movement including receptor dimerization and cell integration should be considered [24]. Binding of G protein was shown to stabilize an activation intermediate in adenosine A_2A_ receptors, and Gsαβγ shifts the receptor equilibrium to a predominantly active ensemble [23]. Furthermore, molecular dynamics simulations have shown that the dynamics of an adenosine A2a receptor vary significantly according to the lipid environment [25].

Motion dynamics are considered to occur in nanosecond to millisecond order. Our DXT measurement adopted recording speed of 100 μs/frame, and nanosecond ordered fast motion could not be captured in this study. Extending analysis including synchronizing timescale with MD simulations may be helpful. Combined analysis with other single molecular analysis techniques, as well as further establish of the DXT studies using reconstituted GPCR-signaling complexes from the purified proteins, will be needed.

## 4. Materials and Methods

### 4.1. DNA Construction and Transfection

A full-length human 5-HT_2A_R was subcloned into pcDNA3.1, and a FLAG sequence (DYKDDDDK) was introduced at the N-terminus using the inverse PCR system (Toyobo biotech, Osaka, Japan). Mutant constructs were made by site-directed mutagenesis using the QuikChange mutagenesis kit (Agilent Technologies, Santa Clara, CA, USA). The sequences of all constructs were verified by DNA sequencing. The FreeStyle HEK 293-F cells (Thermo Fisher, Waltham, MA, USA) were grown under suspension culture in FreeStyle 293 Expression Medium (Thermo Fisher) at 37 °C under 5% CO_2_. Cells were transfected using a mixture at a ratio of one μg plasmid DNA: polyethyleneimine MAX (Polysciences, Warrington, PA, USA) = 1:3 (weight) into 1 mL of cells at a density of 2 × 10^6^ cells/mL. The next day after transfection, the cells were inoculated on the surface of the 12.5 μm thick polyimide films (Kapton, Du Pont-Toray) that were treated with 0.001% poly L-lysine (Peptide institute, Osaka, Japan) at a density of 1 × 10^6^ cells/cm^2^ and further cultured for 12 h. The viability of cells was examined by trypan blue dye exclusion.

### 4.2. DXT Measurements

Gold nanocrystals were fabricated by epitaxial growth on a KCl (100) substrate under 10^−4^ Pa vacuum condition. The gold nanocrystal was modified with the anti-FLAG M2 monoclonal antibody (Sigma-Aldrich: F1804, St. Louis, MO, USA) through thiol-Au chemistry. The gold nanocrystals (20–60 nm diameter) were conjugated with 100 μg FLAG antibody in 1 mL PBS (pH 8.0) under vigorous vortexing. The mixture was further dispersed with ultrasonic sonication for 30 min on ice. Unbound antibody was removed by centrifugation at ×1000 rpm for 10 min twice. The sedimented conjugates were again dispersed in the 1 mL PBS (pH 7.4) and an aliquot of gold nanocrystal solution (50 μL) was reacted with the receptor expressing cell surface (7 × 7 mm area) at room temperature. After 30 min, unbound gold conjugates were washed away twice with PBS. They were covered with 5-μm-thick polyimide films with 20 μL chamber buffer, and sandwiched by stainless steel frames and screw-clamped.

DXT measurements were conducted using the SPring-8 BL40XU beamline. The beam size was adjusted to 50 μm in diameter by inserting a pinhole aperture upstream of the sample, and time-resolved diffraction images from the gold nanocrystals were recorded by an X-ray image intensifier (V7739P, Hamamatsu photonics, Hamamatsu city, Japan) and a CMOS camera (Phantom V2511, Vision Research, Wayne, NJ, USA). For each sample, diffractions at 36 positions (6 × 6) within 1 mm^2^ (1 mm × 1 mm) were recorded. The distance between the sample and the detector was set to 50 mm. X-rays with an energy bandwidth of 0.02 (15 keV in peak energy and photon flux of 10^13^ photon/s) were used for the living-cell DXT to minimize damage to the cells (usual DXT uses X-rays with an energy bandwidth of 0.1 and 15.8 keV in peak energy and photon flux of 10^13^ photon/s).

### 4.3. Image Analysis for DXT

Each diffraction spot was tracked by TrackPy (v0.3.2 https://doi.org/10.5281/zenodo.60550 (accessed on 1 May 2021) after correcting the background. Trajectories were analyzed using a custom software written within IGOR Pro (Wavemetrics, Lake Oswego). Data were recorded with 100 μs per frame and a total measurement time was 10 ms. The number of short-lived data (lifetime ≤ 0.4 ms) were 2.5 times higher than that of the long-lived data (0.5 ≤ lifetime ≤ 10 ms). Therefore the short-lived data were excluded from the analysis for minimizing noise.

Protein dynamics were analyzed using a temporal mean squared displacement (MSD) algorithm to extract the local behavior of the protein as a function of time. The MSD curves were fitted by the following function: *δ*^2^(*t*) = *D**_α_ t^α^* + 2*β*^2^. *D**_α_* is the anomalous diffusion constant, a nonlinear relationship to time. *α* represents subdiffusion (1 > *α* > 0) or superdiffusion (*α* > 1), and *β* is a measurement error.

### 4.4. Immunocytochemistry

The transfected HEK293F cells were cultured for 48 hr on poly-L-lysine coated 24 well plate at a density of 2 × 10^4^ cells per well. Cells transfected with pcDNA3.1 were negative control (mock transfection). The cells were fixed with 4% paraformaldehyde for 30 min, and then permeabilized with 0.05% Triton X-100 in PBS for 10 min. Nonspecific antibody binding was blocked by incubating cells with 1% BSA containing PBS for 30 min. Cells were labeled with anti-FLAG M2 antibody (1 μg/mL) for 1 h, followed by Alexa 555-conjugated secondary antibody (Invitrogen: A31622, 1 μg/mL) for 1 h. Nuclei were stained with DAPI (4’,6-diamidino-2-phenylindole). Cells were examined using an ZEISS Axio Scope.A1 microscope. Pictures were collected by an AxioCam CCD camera (ZEISS) and AxioVision 4.8 software (ZEISS).

### 4.5. Live-Cell Immunofluorescence Assay

All the following steps were performed on ice. The cells in suspension were washed with PBS, precipitated by centrifugation and resuspended in 1 mL PBS containing 0.5% BSA and 1 µg/mL FLAG M2 antibody. After 1 h, cells were washed by centrifugation twice in PBS. They were resuspended in 1 mL PBS containing 0.5% BSA and 1 µg/mL Alexa488-conjugated IgG (Thermofisher: A11001). After 30 min, the cells were washed by centrifugation twice with PBS. The cells were resuspended in 1 mL PBS and dispensed into 96-well plate at a density of 1 × 10^5^ cells per well. Fluorescence signals were measured at Ex/Em 495/519 nm using Flexstation 3 system. Background signals obtained from measurement of nontreated cells were subtracted from all data. Data are presented as mean values ± standard deviation (SD) from four different experiments. Two-tailed paired Student *t*-test *p*-values indicate statistical significance (* *p* < 0.05 and ** *p* < 0.01). 

### 4.6. Calcium Influx Assay

Calcium assay was performed using the FlexStation 3 system with the FLIPR Calcium 6 reagent (Molecular Devices Inc., San Jose, CA, USA). Ketanserin was applied to cells at a final concentration of 10^−5^ to 10^−12^ M at 10 min prior to measurement. Alpha-methylserotonin was applied at 10 μM using the Flex mode. Four parameter logistic curves were generated. Data are presented as mean values ± SD from five different experiments. The concentration at IC50 was analyzed by the Student *t*-test (** *p* < 0.01).

## Figures and Tables

**Figure 1 ijms-22-05285-f001:**
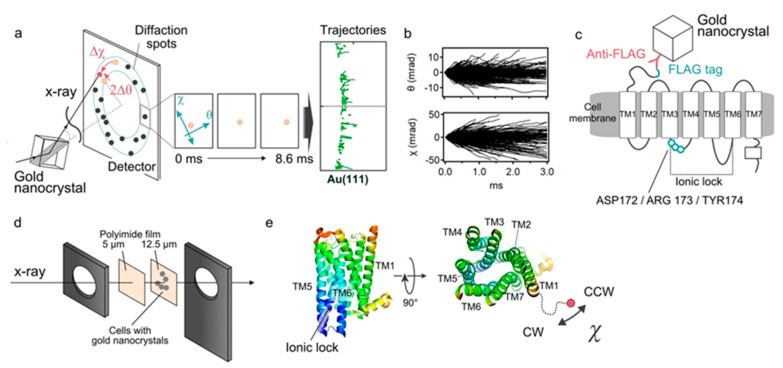
Diffracted X-ray tracking (DXT) measurements. (**a**) Schematic of the DXT measurement. Gold nanocrystals were labeled at specific positions of the target proteins, and their motion was determined by the analysis of the diffraction spots. Pink-beam X-rays elicited trackable diffraction spots from the gold nanocrystals. The central three panels show movement of diffraction spots. The right panel shows piled data of Au (111) spots, representing trajectories. (**b**) An angular displacement along θ and χ axes over time (3.0 ms). The trajectories were projected and analyzed on the χ-θ coordinates, separately. The 1291 traces obtained from the ligand-free condition are presented. (**c**) Labeling of 5-HT_2A_Rs with gold nanocrystals. The FLAG-tag introduced at the N-terminus was labeled by FLAG antibody-gold nanocrystals conjugates. After washing out the unbound gold nanocrystals, the motion of 5-HT_2A_Rs was measured. (**d**) Preparation of the sample holder. FLAG-tagged 5-HT_2A_Rs was transiently expressed in the HEK293 cells, and they were cultured on the surface of 12.5-μm-thick polyimide films. After labeling the receptors with FLAG antibody-gold nanocrystal conjugates, the cells were covered with 5-μm-thick polyimide films with 20 μL chamber buffer. They were sandwiched by stainless steel frames. (**e**) Spatial arrangement of the transmembrane helices of 5-HT_2A_Rs viewed from side and top (reconstituted from the PDB: 6A93). The 5-HT_2A_R comprises intrinsically disordered N-terminal domain of approximately 75 amino acids. Gold nanocrystals were attached to the N-terminal end of the TM1 via the anti-FLAG antibody. Regarding the twisting motion, negative value (Δχ < 0) represent clockwise (CW) rotation, while positive value (Δχ > 0) represent counter-clockwise (CCW) rotation.

**Figure 2 ijms-22-05285-f002:**
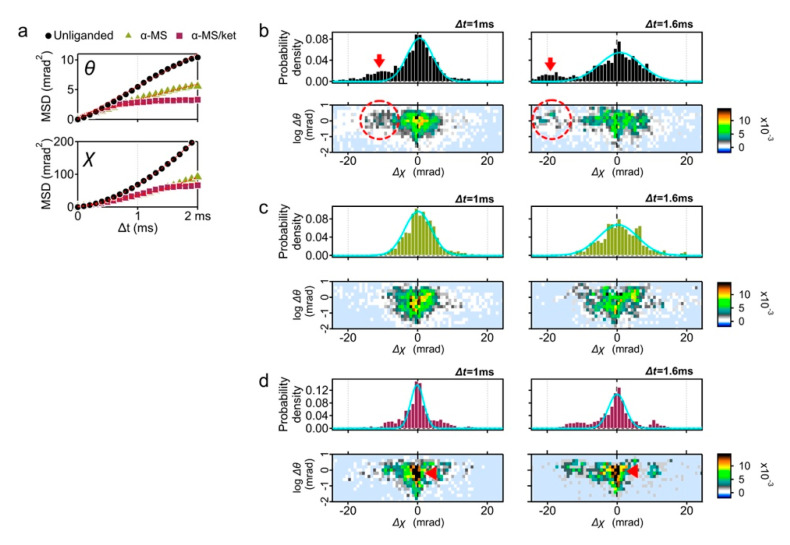
DXT analysis of the wild-type 5-HT_2A_R. (**a**) Mean square displacement (MSD) curves of 5-HT_2A_R motion for the θ and χ axis; unliganded control (black circles), in the presence of 10 μM α-methylserotonin (αMS, green triangles), and 10 μM αMS with 100 μM ketanserin (αMS/ket, red squares). (**b**) Distribution of the absolute angular displacement of the wild-type 5-HT_2A_R over an interval time t (upper left panel, Δt = 1 ms, and upper right panel, Δt = 1.6 ms). Main population in the histograms are fitted by the Gaussian curves. Two-axis distribution maps of θ (vertical axis with logarithmic scale) and χ (horizontal axis) are presented at the lower row. Fast moving component toward CW direction was indicated by arrows (upper panels) and dotted circles (lower panels). The same analysis was applied to the DXT of (**c**) αMS, and (**d**) αMS/ket. Slow moving population in αMS/ket were shown by red arrows.

**Figure 3 ijms-22-05285-f003:**
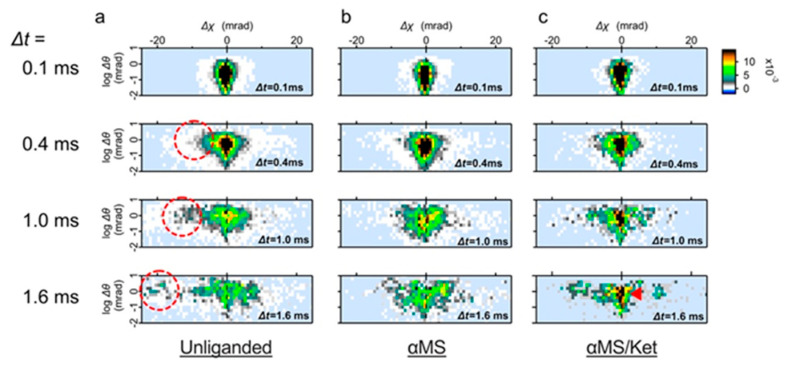
Time course of the wild-type 5-HT_2A_R motion in the two-axis map. (**a**) Distribution transition of the wild type 5-HT_2A_R motion in the absence of ligand. Subpopulation representing twisting motion toward CW (Δχ < 0) are circled. (**b**) Distribution transition of the receptor in the presence of αMS. The fast moving component was not seen. (**c**) Distribution transition of the receptor in the presence of αMS/ket. Low mobility group remained even in the Δt = 1.6 ms (dark filled pixels, indicated by an red arrow).

**Figure 4 ijms-22-05285-f004:**
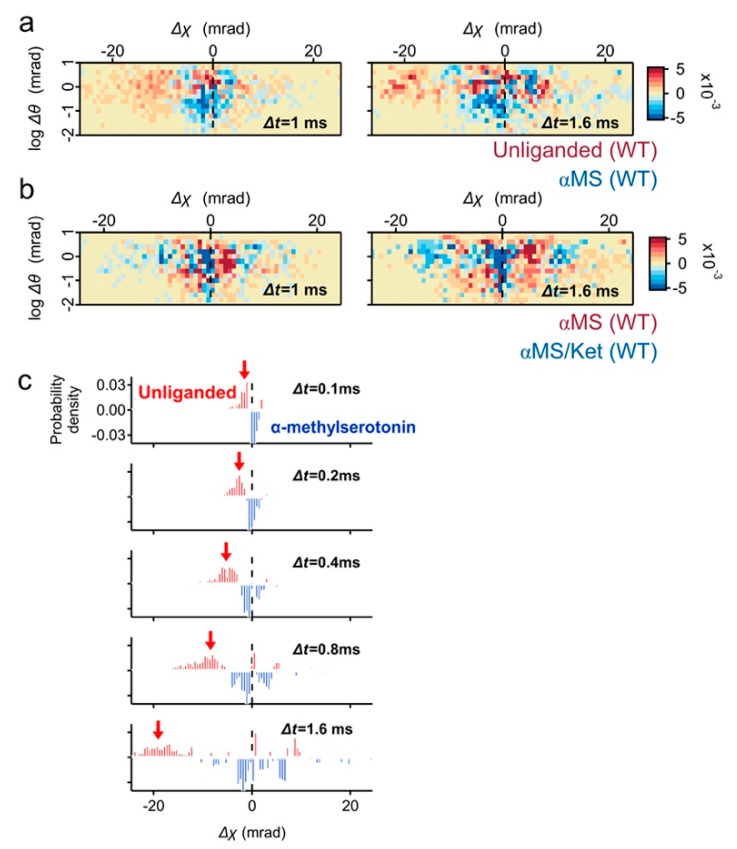
Subtraction analysis of the wild-type 5-HT_2A_R motion in the two-axis map. (**a**) Two-axis subtraction maps of the wild type 5-HT_2A_R between unliganded control (red pixels) and αMS (blue pixels). Population representing CW movement (Δχ < 0) are dominated in the unliganded, while major population of αMS remain at Δχ ~ 0. (**b**) Two-axis subtraction maps of wild type 5-HT_2A_R between αMS (red) and αMS/ket (blue). The large population of αMS/ket remains at Δχ ~ 0. (**c**) Time course of subtraction map in χ distribution between unliganded (blue) and αMS (red). The CW twisting population was clearly shown in unliganded (red arrows).

**Figure 5 ijms-22-05285-f005:**
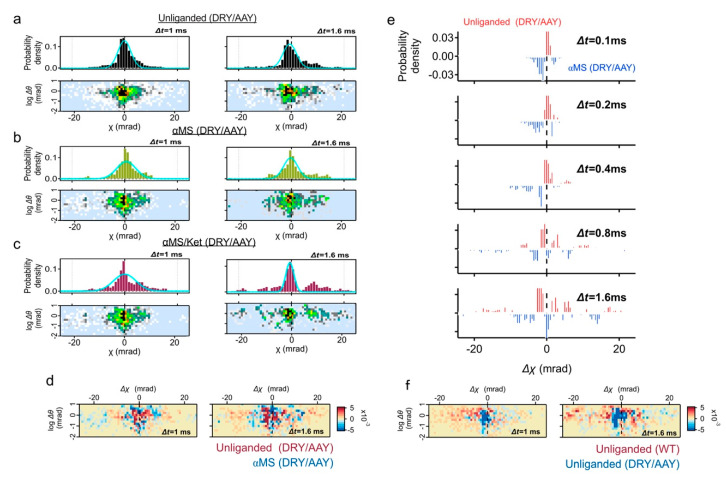
DXT analysis of the DRY/AAY mutant. (**a**–**c**) Distributions of the absolute angular displacement over an interval time t (Δt = 1, and 1.6 ms) for unliganded, αMS and αMS/Ket. Main populations in the histograms are fitted by the Gaussian curves. Two-axis distribution maps of θ (vertical axis with logarithmic scale) and χ (horizontal axis) are presented at the lower rows. (**d**) Two-axis subtraction maps between the unliganded wild-type 5-HT_2A_R (red) and unliganded DRY/AAY mutant (blue). Population representing CW movement (Δχ < 0) are dominated in wild-type 5-HT_2A_R, while low mobility group of DRY/AAY mutant remains at Δχ ~ 0. (**e**) Two-axis subtraction maps of the DRY/AAY mutant between unliganded (blue) and αMS (red). Population of αMS was slightly dispersed. (**f**) Time course of subtraction map in χ distribution of the DRY/AAY mutant between unliganded (blue) and αMS (red).

## Data Availability

The data that support the findings of this study are available from the corresponding author upon reasonable request.

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
