# Peer review of "Living-Cell Diffracted X-ray Tracking Analysis Confirmed Internal Salt Bridge Is Critical for Ligand-Induced Twisting Motion of Serotonin Receptors"

_ijms, 2021, doi:10.3390/ijms22105285_

Round 1
Reviewer 1 Report
In the manuscript “Living-cell diffracted X-ray Tracking Analysis confirmed Internal Salt Bridge is critical for Ligand Induced Twisting Motion of Serotonin Receptors” the authors analyzed intramolecular motions of the 5-HT2AR in whole cells by using diffracted X-ray tracking (DXT). Based on this study, the authors show that the N-terminus of the receptor is highly mobile and undergoes a clockwise twisting movement in the absence of ligands. The fast conformational dynamics of the N-terminus of the receptor were significantly reduced by either full agonist or inverse agonists binding. Furthermore, the authors show that the ionic lock between residues of the D/ERY motif and TM6 is important for the rotational movement of the N-terminus of the receptor. In comparison of a previous study published by the authors on the 5-HT2AR, they were now able to describe the torsional movement of the N-terminal domain of the receptor using the lifetime filtering technique.
While this is an interesting approach to study intramolecular movements of membrane proteins in the native environment of an entire cell, they are certain drawbacks that have to be discussed and that makes the interpretation of the results quite challenging.
Major points:
- The authors overexpressed the receptor in HEK 293 cells. In combination with the usage of an antibody instead of a fab fragment, this could lead to antibody-mediated cross-linking between the receptors at high density. How can this possibility influence the readout of the DXT technique?
- What is the size of the gold nanoparticle used to probe conformational changes of the receptor? Furthermore, the antibody is also pretty big in comparison to the receptor (about three-times the size) – how does the size of the antibody influence receptor dynamics ? Considering the size of the gold nanoparticle and the antibody, the discussed torsional motion of 0.58° at deltat of 1 ms detected for the N-terminus seems to be very small. In addition to that, it is not clear to me how relevant this small motion is for the mechanism of receptor activation. Since this movement is relatively small, it is important to include a control with a truncated N-terminus to see how specific the ligand-dependent effect is on movement of the N-terminus. Alternatively, mutations that disturb the interaction of the N-terminus with the TMs core (analogue to the mutations R6G;E42G for the 5-HT2B receptor) could be used as a good control to see how specific the differences between the different ligand-conditions are.
- How well are the individual diffraction spots from multiple receptors separated? How much can an overlay impact the readout?
- The gaussian fit of the data in Fig. 2 does not the histograms nicely. There seem to be multiple contributions to the distribution. Why did the authors decide to use one gaussian functions and not multiple as done in their previous study? How would this impact the interpretation of the results?
- The authors should explain into more detail what a counterwise rotation means in terms of protein movements – what is causing this directionality? I would expect a relatively flexible N-terminus to move rather randomly when the interaction between the TMs domain and the N-terminus is released.
- What is the cause of the asymmetric displacement distribution of the DRY/AAY mutant ? The authors should also provide data for the DRY/AAY mutant in the presence of aMS/Ket.
- It would be a good addition to the manuscript to also include mutations of the receptor that would stabilize the inactive state and destabilize the active state (e.g. Y380, Y254) to see if the N-terminus in such a mutant behaves similarily compared to the WT in the presence of Ket.
- How do intracellular signalling proteins influence the movement of the receptor ? The authors should discuss possible effects of this interaction for the N-terminal movements, because HEK293 cells do contain a certain amount of intracellular heterotrimeric G proteins.
- The quality of the figures is really bad. Sometimes it is even hard to see the labeling of the axis. The resolution of the figures should be significantly increased.
Minor points:
Line 26: Please, add a space between full agonist and alpha-methylserotonin.
Line 42: Change to: “The structures of 5-HT2AR have been solved by crystallography and cryo-electron microscopy.”
Line 68: replace Quik with Quick
Line 183: add space between 10 uM and alphaMS.
Line 211: add space between and and alphaMS
Author Response
Please see the attached PDF file.

Reviewer 2 Report
Authors used X-ray tracking analysis to detect twisting motion in serotonin receptor. Although, this technique presents relatively new and intriguing method how to monitor motions in protein molecules, the overall novelty of the presented conclusions in rather limited.
Crucial control experiment is missing. Even though, sensitivity of the assay to ligands and to mutations in the serotonin receptor indicates that the signal is likely specific, control experiment showing antibody specificity and 5-HT2A expression in HEK cells is necessary (and the same for the mutated receptor).
I also miss information about:
Anti-FLAG antibody (e.g. sigma cat.#)
Antibody concentration/dilution and composition of the labeling solution used.
Description how the antibody/gold conjugation was done.
How big were the gold nanocrystals compared to the size of the receptor? Can neighboring proteins in the cell membrane contribute to the detected signal?
Line 292: I do not think this statement is correct. Stabilizes yes, but how the data supports the consensus that it stabilizes the receptor in inactive conformation?
Line 237 and 251: If authors claim there is significant difference, they should provide p-value and describe how the statistical analysis was performed.
Supplementary Fig. 1: what is “/T” ?
Supplementary tables: “bird”^2 – probably problem with pdf export?
Author Response
Please see the attached PDF file.

Round 2
Reviewer 1 Report
The authors responded to all concerns appropriately. Since the measurements are dependent on a special beamline setup at Spring-8 the additional experiments suggested cannot be provided in a reasonable amount of time. However, the authors now include descriptions of potential drawbacks of their study and proposed to conduct the suggested tests of their method on 5HT2A in future experiments. The manuscript is significantly improved and can be accepted after the following minor revisions:
Line 103: please correct "After 30 min, unbound gold conjugates were washed twice with PBS" to "After 30 min, unbound gold conjugates were washed away twice with PBS" or After 30 min, the cells were washed twice with PBS to remove unbound gold conjugates:
Line 146: "Expression and function of receptor expressing cells were confirmed by the immunofluorescence study and calcium influx assay" - please replace by "Expression and function of the expressed receptor construct were confirmed by immunofluorescence and calcium influx assay"
Line 281: please change to "...shift from structural equilibrium of multiple states of the WT to an stabilized active conformation.."
Line 397: please change to :"Because the ionic lock stabilizes the structure in an inactive form, its disruption frequently leads to an constitutively active receptor."
Line 429: In cells, it is not clear what stabilizes the high affinity state of the receptor. It is most likely a nucleotide-free state. Therefore the authors should remove the GDP and just mention Gsαβγ alone.
Supplementary figure 1: it would be good to improve the data of the calcium assay. The baseline of the WT at lower concentration is not reached (even lower concentrations should be used) and the standard deviations (?) for the DRY/AAY mutant are quite large. The authors should also add to the figure legend how they generated the data (three biological repeats or three technical repeats ?). Furthermore, they should describe how the error bars were generated (SEM or SD?).
Author Response
Response to Reviewer 1
Point 1:
Line 103: please correct "After 30 min, unbound gold conjugates were washed twice with PBS" to "After 30 min, unbound gold conjugates were washed away twice with PBS" or After 30 min, the cells were washed twice with PBS to remove unbound gold conjugates:
Line 146: "Expression and function of receptor expressing cells were confirmed by the immunofluorescence study and calcium influx assay" - please replace by "Expression and function of the expressed receptor construct were confirmed by immunofluorescence and calcium influx assay"
Line 281: please change to "...shift from structural equilibrium of multiple states of the WT to an stabilized active conformation.."
Line 397: please change to :"Because the ionic lock stabilizes the structure in an inactive form, its disruption frequently leads to an constitutively active receptor."
Line 429: In cells, it is not clear what stabilizes the high affinity state of the receptor. It is most likely a nucleotide-free state. Therefore the authors should remove the GDP and just mention Gsαβγ alone.
Response 1:
Thank you for reading our manuscript carefully and pointing out any careless mistakes. They were all corrected as suggested by the reviewer.
Point 2:
Supplementary figure 1: it would be good to improve the data of the calcium assay. The baseline of the WT at lower concentration is not reached (even lower concentrations should be used) and the standard deviations (?) for the DRY/AAY mutant are quite large. The authors should also add to the figure legend how they generated the data (three biological repeats or three technical repeats ?). Furthermore, they should describe how the error bars were generated (SEM or SD?).
Response 2: We reassessed the inhibition of α-MS induced calcium influx by ketanserin, and added it as a new supplementary figure 1c. In this experiment, we obtained data at lower concentration up to 10-12M, and could obtain flat baseline. The IC50 concentration of ketanserin was 1.05 ±0.57 nM for WT and 4.59 ± 2.79 nM for AAY (mean values ± standard deviation from four different experiments). The value of IC50 was described in the figure.
Method for the calcium influx assay, as well as immunocytochemistry and live-cell immunofluorescence assay, have been added in the Methods section of the manuscript Line 138-167, Page3.

Reviewer 2 Report
The control experiments provided, by the authors (Fig.1 Expression and function of 5-HT2A receptors) do not provide convincing results. The control Western blotting experiment presented in the previous paper (Mio et al., Biochem Biophys Res Commun, 2020) is not relevant to the experiment presented in this manuscript.
Immunofluorescence:
Many of the cells are dead and show Alexa 555 signal, most likely, due to membrane disintegration. Is expression of 5-HT2A toxic? If so, control experiment with 5-HT2A with no FLAG-tag should be also presented. How nontransfected cells looks like? Labeled nontransfected cells should be included as a control. What were the antibodies used for this experiment?
Is this widefield fluorescence microcopy? To prove surface expression and specific immunofluorescence labeling of only surface 5-HT2A receptors, confocal microscopy should be used. Now it is difficult to judge whether the weak Alexa 555 signal is from surface receptors, labeled internalized receptors or internalized Alexa 555 alone.
Inhibition of α-MS induced calcium influx by ketanserin:
The IC50+-error should be included in the figure or figure legend, including statistical test to show whether there is any statistical difference between WT and DRY/AAY.
Beside this control experiment authors have responded to my questions and improved the manuscript adequately.
Author Response
Response to Reviewer 2
Point 1: Immunofluorescence:
Many of the cells are dead and show Alexa 555 signal, most likely, due to membrane disintegration. Is expression of 5-HT2A toxic? If so, control experiment with 5-HT2A with no FLAG-tag should be also presented. How nontransfected cells looks like? Labeled nontransfected cells should be included as a control. What were the antibodies used for this experiment?
Is this widefield fluorescence microcopy? To prove surface expression and specific immunofluorescence labeling of only surface 5-HT2A receptors, confocal microscopy should be used. Now it is difficult to judge whether the weak Alexa 555 signal is from surface receptors, labeled internalized receptors or internalized Alexa 555 alone.
Response 1: We would like to thank the reviewer’s concern regarding the Alexa555 signal in immunocytochemical staining. The low frequency of Alexa 555 positive cells is due to the expression system adopted. This is a transient expression system, and transfection efficiency in HEK293 cells using polyethyleneimine (PEI) has been reported to range from 30% to 70% depending on transfection conditions (1). Considering that about 40% of the cells were positive for Alexa 555 in this experiment (including faintly stained cells), our condition could be further improved. We added the data for mock transfection in supplementary figure 1a, in which the Alexa 555 positive cells were not observed.
Neither PEI transfection treatment nor 5-HT2AR expression was toxic to cells. We tested viability of cells 48 hr after transfection using trypan blue dye exclusion method. Cell viability under all experimental conditions was as high as 97% (97.64 ± 0.44% for WT, 97.71 ± 0.41% for DRY/AAY, and 97.80 ± 0.44% for mock transfection). The data represent mean ± SD from five different experiments.
As the reviewer commented, supplementary figure 1a (although it shows the expressing of 5-HT2AR in transfected HEK-293 cells) did not provide information about surface expression of the receptor. Because confocal microscopy is not available in our research facility, we applied a live‐cell based immunofluorescence assay to address this issue. The transfected cells were labelled with anti-FLAG M2 antibody (Sigma-Aldrich: F1804), followed by Alexa488-conjugated secondary antibody (Thermofisher: A11001). The cells were dispensed into a 96 well plate at a density of 1 × 105 cells per well, then fluorescence intensity was measured. The signals of both WT and DRY/AAY were significantly higher than that of mock transfected cells, suggesting expression of 5-HT2A receptors on the cell surface. The results of the live‐cell immunofluorescence assay have been added to a new supplementary figure 1b.
We added the following sentences in the manuscript Line 178-182, Page 4.
“Expression and function of the expressed receptor constructs were confirmed by the immunocytochemistry, live-cell immunofluorescence assay and calcium influx assay (Supplementary Fig. 1). We also confirmed using the trypan blue dye exclusion method that the cell viability under all experimental conditions was as high as 97% (data not shown).”.
Methods for immunocytochemistry and live‐cell immunofluorescence assay have been added to the Methods section of the manuscript Line 138-160, Page 3.
- de Los Milagros Bassani Molinas M, Beer C, Hesse F, Wirth M, Wagner R. Optimizing the transient transfection process of HEK-293 suspension cells for protein production by nucleotide ratio monitoring. Cytotechnology. 2014;66(3):493-514.
Point 2: Inhibition of α-MS induced calcium influx by ketanserin:
The IC50+-error should be included in the figure or figure legend, including statistical test to show whether there is any statistical difference between WT and DRY/AAY.
Response 2: We reassessed the inhibition of α-MS induced calcium influx by ketanserin, and added it as a new supplementary figure 1c. The concentration of ketanserin was 1.05 ±0.57 nM for WT and 4.59 ± 2.79 nM for AAY (mean values ± standard deviation from four different experiments). The value of IC50 was included in the figure. The p-value was calculated to be 0.015 (*p<0.05). This graph has been added as a new supplementary figure 1c.
Method for the calcium influx assay has been added in the Methods section of the manuscript Line 162-167, Page 3.

Round 3
Reviewer 2 Report
The authors have addressed my concerns, and I recommend publication in the journal.